# Soil Erosion vs. Vineyard Productivity: The Case of the Aglianico del Vulture DOC and DOCG Areas (Southern Italy)

**Maurizio Lazzari** [1,*] and **Marco Piccarreta** [2]

1 Institute of Cultural Heritage Sciences, CNR-ISPC, C/da S. Loja, 85050 Tito Scalo, Italy
2 Ministry of Education, Universities and Research, 00144 Rome, Italy; marco.piccarreta@posta.istruzione.it
* Correspondence: maurizio.lazzari@cnr.it; Tel.: +39-3473628307

**Abstract:** Soil erosion in European Mediterranean vineyards is the main impact factor of uncontrolled solute and nutrient transport, degradation of soil structure, and loss of organic matter, which are key controlling factors in grape productivity and quality. The relationship between soil loss and grape productivity in the Aglianico DOC and DOCG areas of Southern Italy has been studied. Erosion values estimated through the RUSLE model were compared with grape productivity from 2011 to 2019. The obtained results show a negative correlation between the two parameters. The amounts of soil loss for different vineyard slope classes were also considered. The erosion values increase by an order of magnitude moving from the gentle slopes (0–5°) to the steeper ones (>30°), typical of hilly and mountainous areas. The planned shift in the future of vineyards towards these altitudinal bands could prove to be uneconomical if conservative techniques are not carried out.

**Keywords:** soil erosion; andosols; cropland degradation; RUSLE; Southern Apennine

## 1. Introduction

The Mediterranean region is recognized as the European area most susceptible to the highest rates of soil erosion, the lowest levels of organic matter, and severe salinization problems [1]. The combined action of anthropogenic pressures and recent global warming is causing several soils in Europe's Mediterranean region to reach "critical limits to their ability to provide ecosystem services", including agriculture and carbon sequestration. The situation is quite critical since these Mediterranean countries are major producers of wines, olives, nuts, and tomatoes. It is therefore crucial to understand the effects of soil erosion on agricultural production in order to better define adaptive strategies.

Geomorphological, climatic, and edaphic conditions, together with anthropogenic factors, make vineyards in Mediterranean ecosystems prone to soil degradation and erosion [2–4]. About half of the world's vineyard surface is cultivated in Europe (33,000 km$^2$), where more than 70% is concentrated in Italy (6950 km$^2$), Spain (9670 km$^2$), and France (7870 km$^2$) [5]. In addition, soils play a key role within the viticultural agroecosystem, as their structure and mineral supply to plants also influence grape quality and productivity [6]. Recent studies on the impact of climate changes and cover crop soil management in Mediterranean vineyards have clearly shown that increased sediment transport creates several problems related to uncontrolled solute and nutrient transport [7–13], soil structure degradation, and organic matter loss [3,14–20]. As pointed out by [4,12], the study of factors and rates of soil erosion processes under natural conditions is necessary in order to better evaluate soil conservation methods to achieve vineyard sustainability and high grape quality. Some questions arise. Is there a link between negative yield trends and soil erosion? If the two phenomena are linked, what should be expected in the future?

In this paper, we examine the relationship between soil erosion and the productivity of vineyards, considering the case study of Aglianico del Vulture DOC and DOCG in Basilicata, Southern Italy, a region which is very sensitive to environmental changes due

to climatic changes [21,22]. DOC means Controlled Designation of Origin, while DOCG means Controlled and Guaranteed Designation of Origin. This means that wines with DOCG designation are in a higher category than DOC because they are subject to stricter controls than DOC wines to ensure their high quality and authenticity.

Here, we apply a soil erosion model from 2011 to 2019 to study the average soil loss and the productivity of grapes relating to the previously mentioned vintages. Among the different soil erosion models, we have selected the Revised Universal Soil Loss Equation (RUSLE) model [23], which has been applied in several other studies on Mediterranean vineyards [24–28], giving satisfactory results in terms of soil erosion estimates and distribution. Unlike the previous works, the annual erosion is not considered here, but the potential soil loss from November 1st to June 30th of the following year. This choice follows the phenological phases of the vine and its need to absorb nutrients at certain times of the year. The cycle of development and absorption of the root system has two periods of intense activity. The first period coincides with ripening and leaf fall (about the end of October). The second period is between budding and flowering (May–early July). Therefore, the loss of nutrients during the summer season should not be a limiting factor in relation to productivity.

Another important question derives from the recent increase in minimum and maximum temperatures found by [29,30]. The increase in spring temperatures is not only decisive for the early flowering of the vines but could also imply or shift the crops towards higher altitudes. Could the shifting of vineyard crops from the hilly to the mountainous belt be worthwhile? What could happen to the level of erosion? In order to answer these questions, we also evaluated the soil loss in the different altimetric bands (plain from 0 to 300 m, hilly from 300 to 600 m, and mountainous above 600 m) and the different slope classes, according to the classification proposed by the CORINE methodology [31]. An interesting relationship between the two parameters seems to emerge from the results. Higher values of soil erosion correspond to lower grape production, whereas in conditions of less erosion, grape productivity increases.

## 2. Materials and Methods

### 2.1. Study Area and Soils

The study area is geologically located along the external thrust front of the southern Apennine chain, characterized by high uplift rates [30] and very accelerated morpho-evolutionary dynamics, which favor linear erosion phenomena (rill and gully erosion and badlands) and areal (landslides) [32–34], which clearly interact with crops and arable lands. Moreover, geographically it is defined by the Aglianico del Vulture DOC and DOCG wine production region, which spans 1330 km$^2$ in the northeast sector of the Basilicata region, Southern Italy. The soils in the study sector are andosols generated by the pedogenesis of effusive volcanic rocks, characterized by little-evolved profiles at the highest altitudes and more differentiated profiles at the lowest altitudes, with a low apparent density, high water retention capacity, and high cation-exchange capacity. Corresponding to the vineyards, there are mollisols and alkaline soils, well-drained and with high permeability, characterized by a dark red color (browning process), with abundant organic substance in the surface horizons (>1.5%).

Vineyard cropland presently occupies 24.22 km$^2$ of the DOCG area, representing one of the most diffuse cultivations (Figure 1). The landscape elevations range from 100 to 1325 m a.s.l., mainly dominated by 65% of hilly terrain, with 20% of alluvial plain, and 15% mountainous. Vineyards are mainly located in the hilly areas (75%) and, subordinately, in the plains (21%). Only 4% of the vineyards are located in the mountains.

By following the slope classification of the CORINE methodology [31], it is possible to observe (Table 1) that the vineyards cultivated in the plains are mainly set on gentle to flat surfaces (88.4%) and, subordinately, on gentle surfaces (10.5%). The vineyard crops on the hills are also mainly set on gentle to flat surfaces (62.7%) and, subordinately, on gentle surfaces (35%), whereas only 2.5% of them are located on steep slopes. In the mountainous

areas, the vineyards are mainly cultivated on gentle slopes (68.3%), subordinately on gentle to flat slopes (23%), and on steep slopes (8.5%).

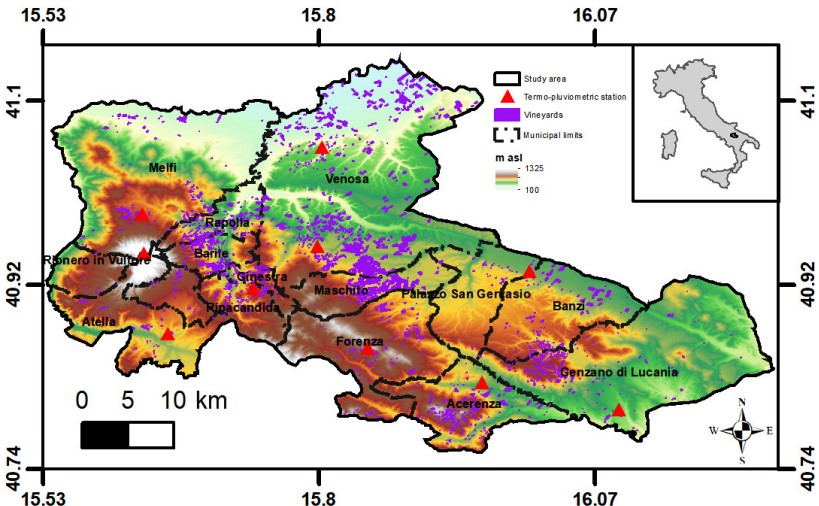

**Figure 1.** Geographical setting of the Aglianico del Vulture area and vineyard distribution.

**Table 1.** Slope class distribution related to altimetric range.

| | Plain | | Hill | | Mountain | |
|---|---|---|---|---|---|---|
| **Slope Classes (°)** | **km²** | **%** | **km²** | **%** | **km²** | **%** |
| gentle to flat (0–5) | 4.39 | 89.31 | 11.17 | 62.76 | 0.27 | 23.06 |
| gentle (5–15) | 0.52 | 10.58 | 6.23 | 35.00 | 0.80 | 68.32 |
| steep (15–30) | 0.005 | 0.10 | 0.39 | 2.19 | 0.10 | 8.54 |
| very steep (>30) | 0.0004 | 0.01 | 0.008 | 0.045 | 0.001 | 0.085 |

In the study area, 10 thermo-pluviometric stations are present. Temperature and precipitation data have been provided with hourly temporal resolution for the last fifteen years. The mean annual temperatures range from around 13 °C to 15 °C, and both minimum (Tmin) and maximum (Tmax) temperatures are increasing [29,30] (Figure 2). The rainfall patterns are strongly conditioned by the geography. According to [35], the annual and seasonal total precipitation underwent a general downward trend over the period 1951–2010. The negative trend is controlled by the fall–winter precipitation collapse. However, the trend in recent years has been positive.

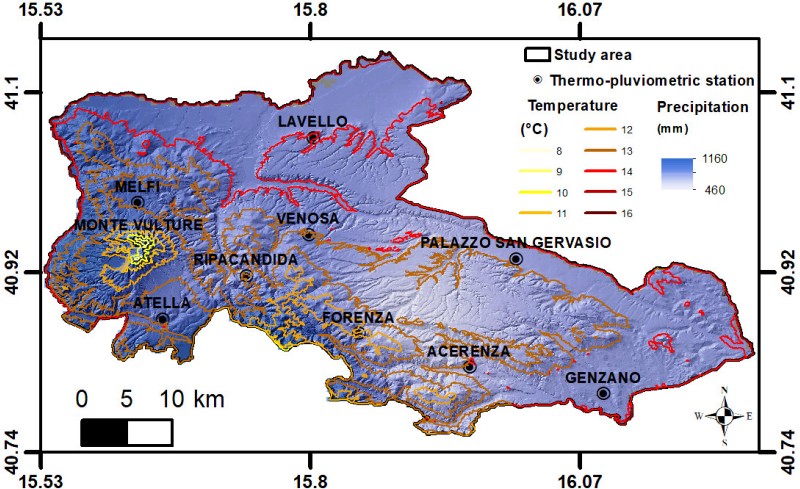

**Figure 2.** Mean temperature and mean precipitation in the study area.

### 2.2. Grape Productivity Data

The grape productivity data for the Aglianico DOC and DOCG from 2011 to 2019 were collected from the Italian National Institute of Statistics (ISTAT), which provides the amounts of grapes harvested for wine production (in quintals) and the extent of the vineyards (in hectares) in freely available yearly publications in Italy.

The ISTAT grape productivity, here defined as grape yield (q) per hectare of vineyard, was used to investigate the impact of potential soil erosion on wine production.

### 2.3. RUSLE Model

The RUSLE model estimates average annual erosion using rainfall, soil, topography, and management using the following equation:

$$A = R \times K \times (LS) \times C \times P \tag{1}$$

where A is the computed site soil loss ($Mg \cdot ha^{-1} \cdot year^{-1}$); R is the rainfall erosivity factor ($MJ \cdot mm \cdot ha^{-1} \cdot h^{-1} \cdot year^{-1}$); K is the soil erodability factor ($Mg \cdot ha \cdot MJ^{-1} \cdot mm^{-1}$); L is the slope length factor that is often combined with S, a slope steepness factor, to yield a unitless terrain factor (LS); C is the unitless vegetation cover factor; and P is the dimensionless erosion control practice factor.

#### 2.3.1. Rainfall Erosivity

Rainfall erosivity measures rainfall's kinetic energy (E) and intensity to describe the effect of rainfall on sheet and rill erosion. Its calculation requires precipitation data at short time intervals (1–60 min). The R factor is calculated by multiplying E by the maximum rainfall intensity over a period of 30 min ($I_{30}$) for each rainstorm. It accumulates the rainfall erosivity of individual rainstorm events, averages this value over multiple years, and is calculated as:

$$R = \frac{1}{n} \sum_{j=1}^{n} \sum_{k=1}^{mj} (EI_{30})_k \tag{2}$$

where R ($MJ \cdot mm \cdot ha^{-1} \cdot h^{-1}$) is the average annual R factor, n is the number of years in the record, $m_j$ is the number of erosive events during year j, and $EI_{30}$ ($MJ \cdot mm \cdot ha^{-1} \cdot h^{-1}$) is the rainfall erosivity for event k. The erosivity of a particular event is

$$EI_{30} = \left( \sum_{r=1}^{m} e_r \cdot v_r \right) \cdot I_{30} \tag{3}$$

where $e_r$ is the kinetic energy per unit of rainfall ($MJ \cdot ha^{-1} \cdot mm^{-1}$); $v_r$ is the rainfall depth (mm) for the time interval, r, of the hyetograph, which has been divided into r = 1, 2, ..., m subintervals; and $I_{30}$ is the maximum rainfall intensity over a 30 min duration. The rainfall kinetic energy, $e_r$, is calculated with the equation proposed by [36], where i is the rainfall intensity ($mm \cdot h^{-1}$) at each given time interval:

$$e_r = 0.29 \left( 1 - 0.72 e^{-0.05i} \right) \tag{4}$$

A rainfall event is divided into two parts if its cumulative depth for a duration of 6 h at a certain location is less than 1.27 mm. According to the authors' guide [37], rainfall is considered erosive if it has a cumulative value greater than 12.7 mm. Here, we assume as erosive a rainfall event > 8 mm, a threshold which is considered more suitable due to the prevalence of loose lithologies. With reference to what is reported in the Introduction, we consider erosion values from 1 November to 30 June of the following year as annual values. The R factor has been calculated from hourly data by using the Rainfall Intensity Summarisation Tool (RIST 3.99.12) software [38]. This is the lowest time resolution available and, unfortunately, produces an underestimation of the real calculation by more than 50% [39].

### 2.3.2. Soil Erodibility

The soil erodibility represents both the susceptibility of soil to erosion and the rate of runoff, as measured under the standard unit plot condition. The soil erodibility factor (K) map has been calculated by using the USLE equation [40]:

$$K = [2.1 \times 10^{-4} (12 - M) [(Si + fS)(100 - C)] \ 1.14 + 3.25(A - 2) + 2.5(P - 3))]/100 \quad (5)$$

where M is the organic matter content (%); Si is the silt content (%), 2 to 50 μm; fS is the fine sand content (%), 50 μ to100 μm; C is the clay content (%), less than 2 μm; S is the sand content (%), 50 μm to 2 mm; A is the structure; and P is the permeability class (within top 0.60 m). The soil data were derived from different sources.

The soil map of Basilicata was used to derive texture, structure, and permeability data [40]. The organic matter data were derived from the European Soil Database [41].

### 2.3.3. Slope/Length (LS) Factor

The LS factor was calculated with the equation proposed by [42]:

$$LS = \left( \frac{A}{22.13} \right)^m \left( \frac{sin\ \beta}{0.09} \right)^n \quad (6)$$

where $A$ is the upslope contributing area, $\beta$ is the steepest slope angle (radian), and $m$ and $n$ are the slope length exponent and slope steepness exponent, respectively, whose values depend on the type of flow and soil properties.

The values of exponents range from $m$ = 0.2 to 0.6 and $n$ = 1.0 to 1.3, where lower values are used for prevailing sheet flow and higher values for prevailing rill flow [43].

The Digital Elevation Models of 2013 (scale 1:5.000) from the geoportal of the Basilicata region was used. The TAUDEM extension in ArcGIS 10.5 was used to compute the upslope contributing area, as it employs the D-∞ algorithms [44].

### 2.3.4. Cover Factor

The cover and management factor (C) reflects the effect of cropping and management practices on the soil erosion rate. Generally, the C factor ranges between 1 and 0, where values equal to 1 indicate no cover present (barren land), and values near zero (0) indicate very strong cover effects and well-protected soil. Land use data derived from the geoportal of the Basilicata region were used to calculate the C values by following the authors' suggestions [37]. In particular, a value of 0.27 was assigned for vineyards (Table 2).

**Table 2.** Cover factor values of the recognized types of cover.

| Type of Coperture | Cfactor Value | Type of Cover | Cfactor Value | Type of Cover | Cfactor Value |
|---|---|---|---|---|---|
| Water bodies | 0 | Pastures | 0.004 | Crops | 0.2 |
| Urban areas | 0 | Shrubs | 0.003 | Vineyards | 0.27 |
| Scrublands | 0.1 | Forest | 0.001 | Fruit trees | 0.28 |
| Grassland | 0.009 | Vegetable gardens | 0.3 | Olive groves | 0.35 |
| Land mainly occupied by agriculture, with significant areas of natural vegetation | 0.17 | | | Citrus groves | 0.12 |

### 2.3.5. Support Practice Factor

The support practice factor (P) is the soil loss ratio with a specific support practice to the corresponding soil loss with up- and downslope tillage. Due to the absence of conservation practices on natural slopes, we have adopted a unitary value in this study.

The RUSLE model was applied using ArcGIS 10.5 to obtain 5 m resolution raster maps for each factor.

### 3. Results

#### 3.1. RUSLE Factors Application

The different data inputs processed in ArcGIS 10.5 created four factor maps.

#### 3.1.1. Rainfall Erosivity Computation

The R factor was calculated from hourly data from 2011 to 2019. For each year, the sum of single erosive events from 1 October of the previous year until 30 June of the same year was computed. Nine erosivity maps were created by using the cokriging geostatistical method, as suggested by [45] (Figure 3).

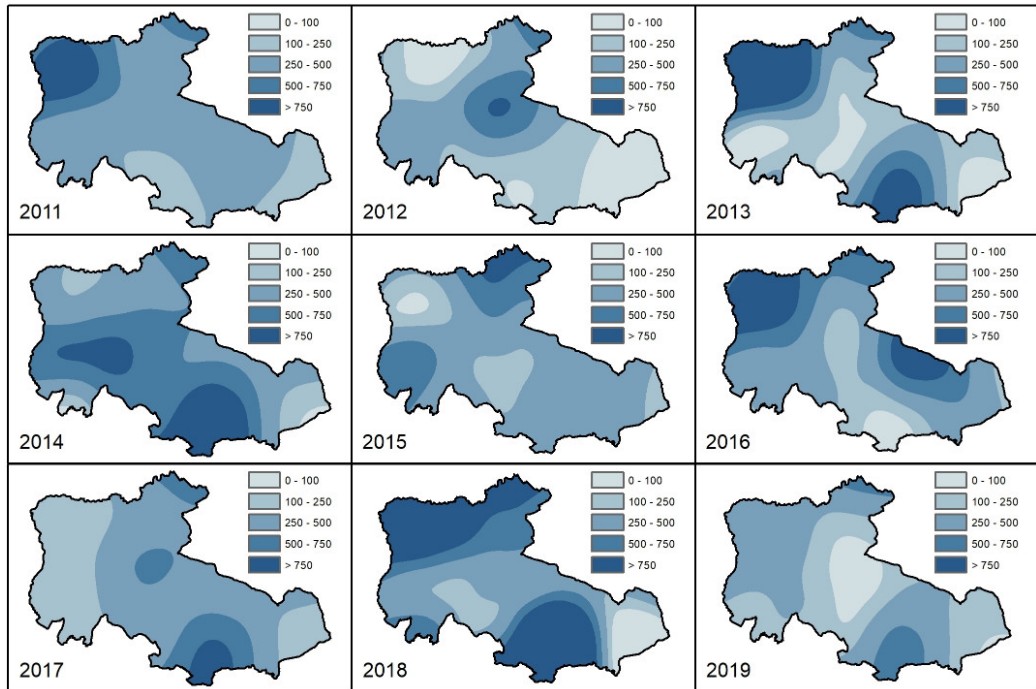

**Figure 3.** Annual rainfall erosivity map (MJ·mm/ha·h) from 2011 to 2019 of the Aglianico del Vulture DOC and DOCG areas.

#### 3.1.2. Soil Erodibility Computation

Equation (5) was applied, and the mapping result is reported in Figure 4a.

#### 3.1.3. Slope/Length Factor Computation

Equation (6) was applied by using the values $m = 0.4$ and $n = 1.2$ because, on gentle vineyards, sheet erosion is the main erosive agent, while on steep slopes, rill erosion dominates over the other erosion processes:

$$LS = \left(\frac{A}{22.13}\right)^{0.4} \left(\frac{sin\ \beta}{0.09}\right)^{1.2} \tag{7}$$

The resulting LS (Length Factor) factor map for Basilicata is reported in Figure 4b.

#### 3.1.4. Cover Factor Computation

Values have been assigned to the different land use typologies. Vineyards, with 0.35, have the highest value of C factor. The C factor maps are reported in Figure 4c.

#### 3.2. RUSLE Model Application

The RUSLE model was applied to the study area (Figure 5).

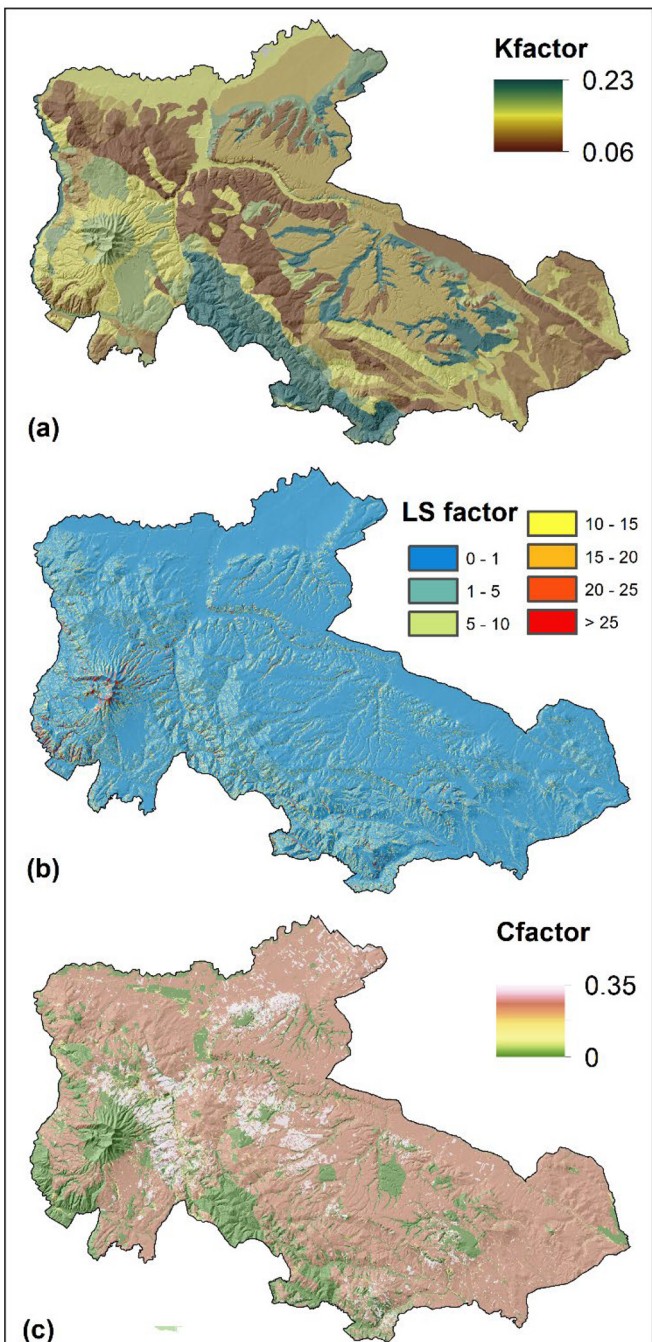

**Figure 4.** (**a**) Soil erodibility map, (**b**) slope/length factor map, and (**c**) cover factor map of the Aglianico del Vulture DOC and DOCG areas.

The results from each raster cell were separated into the following five classes of erosion:

1. Very high erosion ($>40$ Mg ha$^{-1}$ y$^{-1}$);
2. High erosion (10 to 40 Mg ha$^{-1}$ y$^{-1}$);
3. Moderate erosion (5 to 10 Mg ha$^{-1}$ y$^{-1}$);
4. Low erosion (1 to 5 Mg ha$^{-1}$ y$^{-1}$);
5. Very low erosion (0 to 1 Mg ha$^{-1}$ y$^{-1}$).

Through the use of the "extracting by mask" function in ArcGis 10.5, it was possible to evaluate the erosion quantities for the vineyards. The mean annual soil loss calculated from 2011 to 2019 for the vineyards of the Aglianico del Vulture DOC and DOCG areas is reported in Table 3. In the same table, erosion data for the altimetric range (plain, hills, and

mountains) and the CORINE methodology slope classes (gentle to flat, gentle, steep, and very steep) have been inserted.

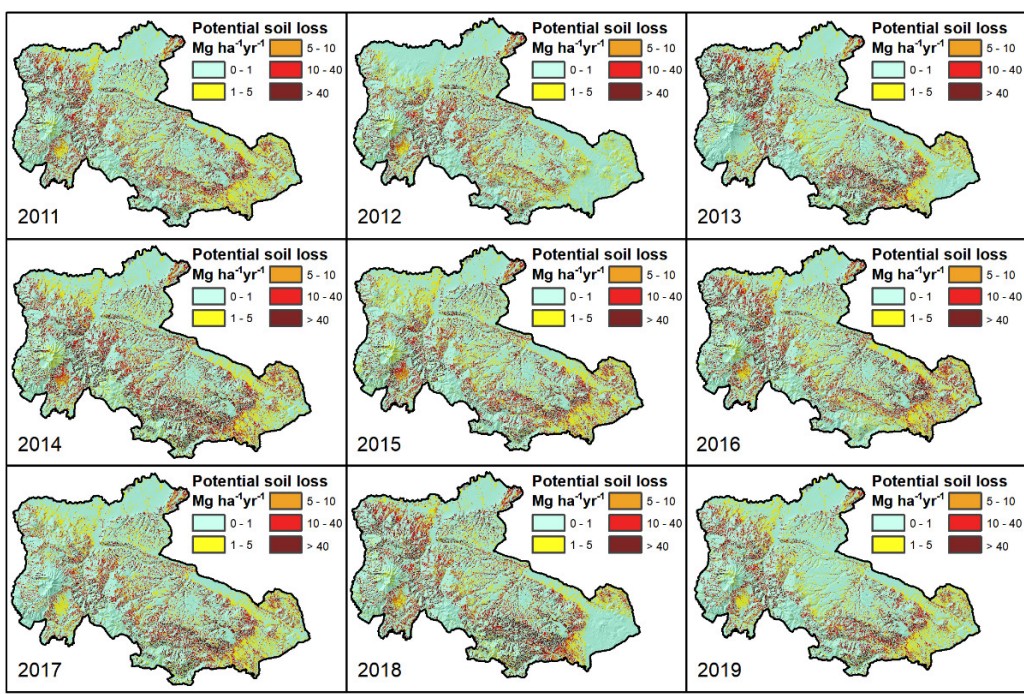

**Figure 5.** Potential soil loss map (Mg·ha$^{-1}$·yr$^{-1}$) of the Aglianico del Vulture DOC and DOCG areas from 2011 to 2019.

**Table 3.** Mean annual soil loss (Mg·ha$^{-1}$) from 2011 to 2019 in the vineyards of Aglianico del Vulture DOC and DOCG areas also related to altimetric range and slope classes.

|  | **2011** | **2012** | **2013** | **2014** | **2015** | **2016** | **2017** | **2018** | **2019** |
|---|---|---|---|---|---|---|---|---|---|
| Vineyards | 6.19 | 6.28 | 6.57 | 13.20 | 6.76 | 6.43 | 7.85 | 11.35 | 4.92 |
| Plain | 2.44 | 2.11 | 2.85 | 2.66 | 3.00 | 2.82 | 2.16 | 4.62 | 1.49 |
| Hill | 5.78 | 6.54 | 5.33 | 11.87 | 5.86 | 5.75 | 7.55 | 8.10 | 3.98 |
| Mountain | 14.57 | 11.62 | 23.95 | 48.51 | 20.25 | 12.93 | 28.06 | 58.52 | 19.68 |
| Gentle to flat slopes | 2.31 | 2.70 | 1.98 | 3.91 | 2.19 | 2.31 | 2.66 | 3.26 | 1.28 |
| Gentle slopes | 11.36 | 11.50 | 12.80 | 25.17 | 12.56 | 11.39 | 14.90 | 18.99 | 9.43 |
| Steep slopes | 32.80 | 33.87 | 26.33 | 74.23 | 41.97 | 36.34 | 40.67 | 33.57 | 27.45 |
| Very steep slopes | 42.24 | 34.98 | 56.00 | 47.37 | 28.90 | 43.21 | 24.10 | 108.49 | 21.07 |

The average value of soil loss calculated every year was moderate, but always above the tolerance threshold (1 Mg·ha$^{-1}$). The highest values were recorded in 2014 and 2018 due to highly erosive rain events.

The average values of soil loss with reference to the altitudinal bands were also considered (Figure 6). In the plains, the soil loss is decidedly low, slightly higher than the tolerable loss threshold. Much higher values, as might be expected due to the increase in the slopes, are recorded in the hilly and mountainous areas. The loss of soil in the hilly belt is moderate to slightly high, whereas the loss of soil in the mountainous belt is severe, with values of an order of magnitude higher than those of the plain.

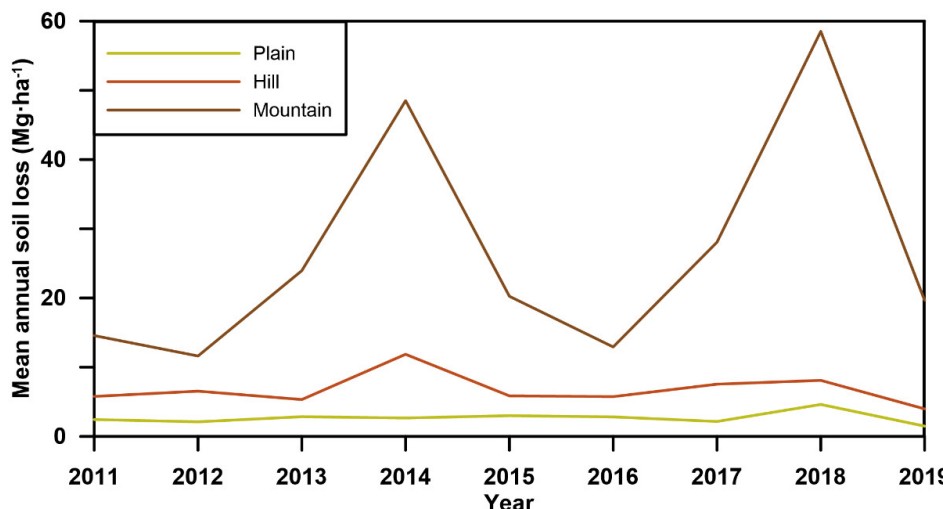

**Figure 6.** Mean annual soil loss (Mg·ha$^{-1}$) in the different altitudinal bands of vineyards of the Aglianico del Vulture DOC and DOCG areas.

The mean values of soil loss were also plotted with reference to the CORINE methodology slope classes (Figure 7). Lower tolerable values of soil loss characterized the gentle to flat slope, whereas, on gentle slopes, vineyards suffer moderate erosion with simulated losses of more than 10 Mg·yr$^{-1}$. High to very high soil losses were recorded on vineyards set on steep slopes, which are more characteristic in the hilly/mountainous belt. It is interesting to note in the graph the contribution of the R factor in the final calculation. In fact, in 2014 and 2017, the potential erosion was higher on steep slopes since, in these two years, the most erosive precipitation (Figure 3) was concentrated in areas characterized by these types of slopes rather than on very steep slopes.

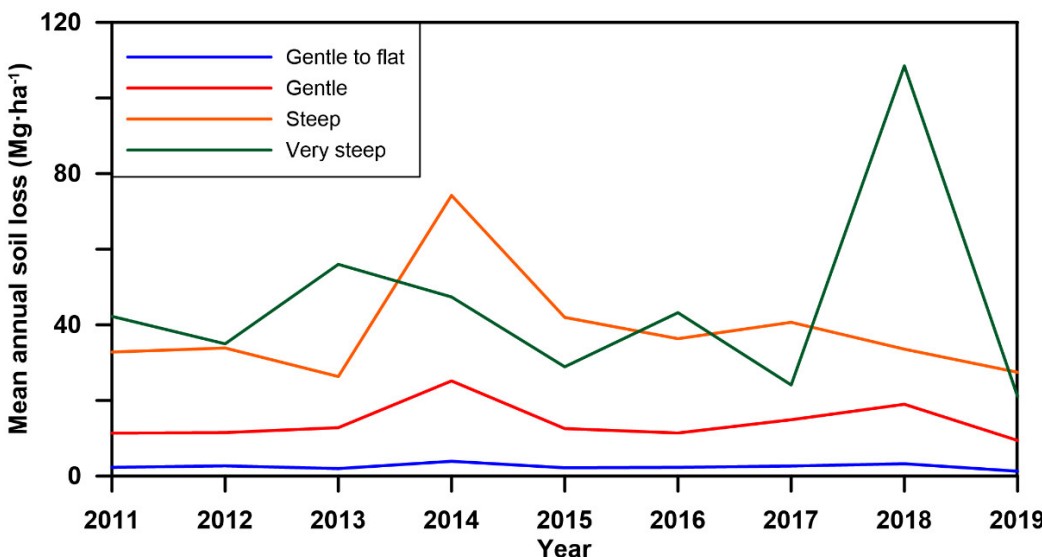

**Figure 7.** Mean annual soil loss (Mg·ha$^{-1}$) in reference to the different slope classes of vineyards of the Aglianico del Vulture DOC and DOCG areas.

### 3.3. Soil Loss Dynamics and Grape Productivity

The annual values of soil loss and productivity from 2011 to 2019 are plotted in Figure 8. The result of the Pearson statistical test [46] is −0.497, which indicates a moderately significant negative correlation.

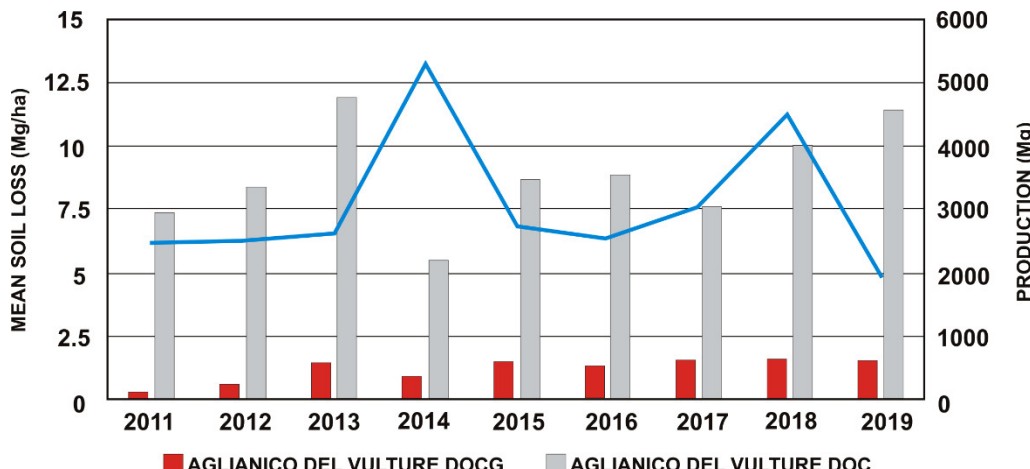

**Figure 8.** Mean annual soil loss (Mg·ha$^{-1}$) in vineyards of the Aglianico del Vulture DOC and DOCG areas versus annual grape production in Mg.

High values of soil loss generally correspond to low productivity values in the vineyards, and vice versa. It should be specified that the only correlation that does not clearly correspond (2018) is linked to very high erosion values recorded on 30 June, i.e., at the end of the annual observation period. This highly erosive episode probably did not affect grape productivity, since the acquisition of nutrients, for the most part, has been carried out, as usual, by the plants. If the 2018 data were excluded from the statistical analysis, the Pearson statistical test would assume a value of −0.71, which, with reference to the altitudinal bands, also indicates a strong correlation between soil loss and grape productivity.

## 4. Discussion

The results of the analyses carried out show statistically significant correlations between soil loss and vineyard productivity, although the small number of years of observation must always be considered. In each year, the estimated soil loss is always higher than the tolerable one. The absence of conservative agricultural practices that limit the losses of nutrients and organic matter, both due to leaching and washing away, seems to exacerbate the decrease in production yields. In a recent paper on steep slope Northern Italy vineyards [47], the authors show that both a single tillage and the use of a mix of nectariferous herbaceous species better mitigates the soil erosion process in comparison with grassed but untilled management. They also judge continuous tillage, which produces the highest concentration of sediment in the runoff, to be untenable in terms of erosion yield. The latter seems to be the most widely used practice in the area of study.

On the other hand, by monitoring runoff and soil losses caused by natural rainfall events over a 12-year period in an experimental vineyard located in Alto Monferrato (NW Italy), Biddoccu et al. [48] found that the protective role of grass cover will be more and more relevant. They measured soil loss from three plots, each of which was managed with a different inter-row soil management practice: conventional tillage (CT), reduced tillage (RT), and controlled grass cover (GC), respectively. The annual average runoff coefficients were 17.4% in CT and 15.3% in RT, while in the GC plot, it was limited to 10.3%. The highest soil losses were observed for the tilled plots, with average yearly erosion rates of 10.4 and 24.8 Mg ha$^{-1}$ year$^{-1}$ in the CT and RT plots. Only 2.3 Mg ha$^{-1}$ year$^{-1}$ was recorded for the GC treatment. The absence of grassland between the rows in the management of vineyards in Basilicata therefore constitutes an aggravation of the loss of soil.

Moreover, in hilly and mountainous areas, the vines are planted in the same direction as the slope, a practice that promotes higher rates of soil erosion [49].

The low yield of the years with the greatest soil erosion cannot be attributed to calamitous events such as extreme droughts or hailstorms, since no information source has ever reported news to this effect. The maximum annual erosion detected can be attributed to

high precipitation, which can also be inferred from the R values. Massano et al. [50] studied the climate impact on grape productivity in Italy by means of the application of bioclimatic indices, which take into account both temperature and precipitation. They found that the relationship between precipitation indices and grape productivity is statistically significant only in Northwest Italy, becoming weaker in Southern Italy. However, they did not use an intensity index such as the power of the rain. Bagagiolo et al. [51], in an experimental study in Northern Italy, found that the highest runoff and soil losses depend on a few extreme rainfall events, namely "long-lasting" (rainfall duration > 50 h) or "intense" events (rainfall Imax30 > 16 mm/h). In particular, the rainfall maximum intensity and rainfall depth were the most important rainfall variables in predicting the degree of runoff and soil loss. The results of this study confirm this last interpretation. The productivity of vineyards in Basilicata is closely related to the power of the rainfall, expressed as erosivity. The increase in short-duration rainfall events of high intensity corresponds to a decrease in grape productivity.

## 5. Conclusions

The relationship between soil loss and grape productivity in a representative area of Southern Italy was studied, although the small number of years of observation must always be considered. The obtained results show a negative correlation between the two parameters, highlighting the impactful role of erosive precipitation and potential soil erosion on grape productivity. High erosion rates, often linked to high rainfall erosivity, correspond to declines in grape production. Conversely, when erosion rates are moderate, the annual value of vineyard productivity increases. Field observations show that the management of the vineyards, allocated mostly on steep slopes, does not involve the use of conservation practices, including, for example, the presence of grassland between the rows. This leads us to hypothesize that the loss of soil, and therefore of organic and nutritious matter during the phenological cycle of the vine, significantly affects the development of the bunches and, as a result, the vineyards' productivity.

The erosion values increase by an order of magnitude moving from the gentle slopes to the steeper ones, typical of hilly and mountainous areas. With a view to the future and foreseeable shift in the vine-growing areas towards high altitudes, the data deserve great attention. The increase in potential erosion and the consequent need to fertilize the land repeatedly to guarantee its fertility will certainly lead to more expensive management of the vineyards. Will production be able to match or exceed these additional costs?

**Author Contributions:** Conceptualization, M.L. and M.P.; methodology, M.L. and M.P.; software, M.P.; validation, M.L. and M.P.; formal analysis, M.L. and M.P.; investigation, M.L. and M.P.; resources, M.L. and M.P.; data curation, M.P.; writing—original draft preparation, M.L. and M.P.; writing—review and editing, M.L. and M.P.; visualization, M.L. and M.P.; supervision, M.L. and M.P.; project administration, M.L. and M.P.; funding acquisition, M.L. All authors have read and agreed to the published version of the manuscript.

**Funding:** This research received no external funding.

**Institutional Review Board Statement:** Not applicable.

**Informed Consent Statement:** Not applicable as not involving humans.

**Data Availability Statement:** The data presented in this study are available on request from the corresponding author.

**Conflicts of Interest:** The authors declare no conflict of interest.

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
