# Peer review of "Soil Erosion vs. Vineyard Productivity: The Case of the Aglianico del Vulture DOC and DOCG Areas (Southern Italy)"

_sustainability, doi:10.3390/su152215700_

Round 1
Reviewer 1 Report
Comments and Suggestions for Authors
This manuscript focuses on Soil erosion in European Mediterranean vineyards. Erosion values estimated through the RUSLE model were compared with grape productivity from 2011 to 2019. The results show a negative correlation between the two parameters. Therefore, The planned shift in the future of vineyards towards these altitudinal bands could prove to be uneconomical if conservative techniques are not carried out. This paper is interesting. However, the manuscript still needs minor revision and check before formal publication.
Here are some specific suggestions for the authors.
1. Figure 1 and Figure 2 are suggested to expressed in longitude and latitude on mapborder. And in the Figure 2, we can not see number 12,13,14,15,16 in the Map body. Also, we suggest that in the map body, the number should be labelled in the middle of the isothermal line. For example, like this, “-----13-------”.
2. In the section “3.1.1. Rainfall erosivity computation”(Line 219), we can see “Nine erosivity maps were created by using the kriging geostatistical method”. However, the kriging geostatistical method has some other different types, for example, it has ordinary kriging, universal kriging, and simple kriging. So which kind of kriging is applied in this study? And there is some other spatial interpolation such as IDW interpolation, so why do authors only select kriging geostatistical method? Different geostatistical interpolation method can have different maps.
3. Authors must tell readers what is the cell size about all the raster map, include Rainfall erosivity, Soil erodibility, Cover factor, Slope/Length and Support practice factor. We believe that various factors coverage normally have the same cell size when overlay is conducted.
4. In Figure 7 the line 278, why the mean anual soil loss in very steep is lower than that steep in 2014-2015? Please explain it.
5. What is the dtat resource which net URL of annual grape production from 2011 to 2019 (Figure 8), which should be explained in the Method section.
6. (Line-155) The “er” should be “er”. Please check it.
7. (Line-209) In the Section “2.1.5. Support practice factor”, There is a sentence “Due to the absence of conservation practices on natural slopes, we have adopted a unitary value in this study”. So how much is the unitary value? How to produce the value? Is there any reference to support the value?
Comments on the Quality of English LanguageGood
Author Response
This manuscript focuses on Soil erosion in European Mediterranean vineyards. Erosion values estimated through the RUSLE model were compared with grape productivity from 2011 to 2019. The results show a negative correlation between the two parameters. Therefore, the planned shift in the future of vineyards towards these altitudinal bands could prove to be uneconomical if conservative techniques are not carried out. This paper is interesting. However, the manuscript still needs minor revision and check before formal publication.
Here are some specific suggestions for the authors.
- Figure 1 and Figure 2 are suggested to expressed in longitude and latitude on mapborder. And in the Figure 2, we can not see number 12,13,14,15,16 in the Map body. Also, we suggest that in the map body, the number should be labelled in the middle of the isothermal line. For example, like this, “-----13-------”.
Done
- In the section “3.1.1. Rainfall erosivity computation”(Line 219), we can see “Nine erosivity maps were created by using the kriging geostatistical method”. However, the kriging geostatistical method has some other different types, for example, it has ordinary kriging, universal kriging, and simple kriging. So which kind of kriging is applied in this study? And there is some other spatial interpolation such as IDW interpolation, so why do authors only select kriging geostatistical method? Different geostatistical interpolation method can have different maps.
Thanks for the clarification request. We considered that, as suggested by Govaerts (1990), the cokriging interpolation methods, together with the use also of elevation data, allows to built more accurate rainfall erosivity maps.
- Authors must tell readers what is the cell size about all the raster map, include Rainfall erosivity, Soil erodibility, Cover factor, Slope/Length and Support practice factor. We believe that various factors coverage normally have the same cell size when overlay is conducted.
Cell size is 5 meter
- In Figure 7 the line 278, why the mean annual soil loss in very steep is lower than that steep in 2014-2015? Please explain it.
Done
- What is the dtat resource which net URL of annual grape production from 2011 to 2019 (Figure 8), which should be explained in the Method section.
Done
- (Line-155) The “er” should be “e ”. Please check it.
Done
- (Line-209) In the Section “2.1.5. Support practice factor”, There is a sentence “Due to the absence of conservation practices on natural slopes, we have adopted a unitary value in this study”. So how much is the unitary value? How to produce the value? Is there any reference to support the value?
R. Normally, the absence of conservation practices on a slope is an aggravation for the intensification of erosion phenomena. Therefore, it constitutes the highest risk of erosion at the management level. This gives rise to the maximum risk value, which is one.
Reviewer 2 Report
Comments and Suggestions for Authors
Line 67: please write elevation ranges that you considered for "different altimetric bands" and data source for this.
Please include the locations of 10 termo-pluviometric stations on Figure 1.
Please write cover values of land cover types that you used in this study under Section 2.1.4. Cover factor"
Figure 4 is not clears Sub-figures should be bigger.
Authors should include a Discussion Section in which they should compare their results with the similar studies in the literature. They need to also discuss advantages and disadvantages.
Conclusion is repeating the results. Please give a concrete scientific message and some future insights.
Comments on the Quality of English LanguageMinor editing of English language required
Author Response
Line 67: please write elevation ranges that you considered for "different altimetric bands" and data source for this.
Done
Please include the locations of 10 termo-pluviometric stations on Figure 1.
Done
Please write cover values of land cover types that you used in this study under Section 2.1.4. Cover factor"
Done
Figure 4 is not clears Sub-figures should be bigger.
Figure has been redrawn
Authors should include a Discussion Section in which they should compare their results with the similar studies in the literature. They need to also discuss advantages and disadvantages.
The referee is right. We have added a Discussion Section
Conclusion is repeating the results. Please give a concrete scientific message and some future insights.
Conclusion has been rewritten
Reviewer 3 Report
Comments and Suggestions for Authors
Sustainability-2670137
Soil erosion vs vineyards productivity: the case of the Aglianico del Vulture DOC and DOCG areas (southern Italy)
This is an article about a practical example of applying RUSLE to evaluate soil loss in vineyards.
The methodology used is not new but it is correct, and the results obtained and discussion are satisfactory. Final remarks are adjusted to the results obtained.
However, the study presented does not represent a significant advance in the understanding of erosion mechanisms of the soil nor for the evaluation of its susceptibility.
Authors should define the meaning of the acronyms DOC and DOCG in the introduction of the manuscript. The terms soil loss and soil erosion appear in the keywords. However, throughout the manuscript it appears that both terms are used interchangeably. It would be appropriate for the authors to define these two concepts and clarify whether they consider them synonymous or not.
Author Response
This is an article about a practical example of applying RUSLE to evaluate soil loss in vineyards. The methodology used is not new but it is correct, and the results obtained and discussion are satisfactory. Final remarks are adjusted to the results obtained.
However, the study presented does not represent a significant advance in the understanding of erosion mechanisms of the soil nor for the evaluation of its susceptibility.
In fact, the objective of this work is another, namely to examine the relationships between soil erosion and the productivity of the vineyards
Authors should define the meaning of the acronyms DOC and DOCG in the introduction of the manuscript.
Done
The terms soil loss and soil erosion appear in the keywords.
we changed the keyword
However, throughout the manuscript it appears that both terms are used interchangeably. It would be appropriate for the authors to define these two concepts and clarify whether they consider them synonymous or not.
Soil erosion and soil loss are similar concepts that refer to the removal of soil from one location and its transportation to another location. But, there is a subtle difference between the two terms.
In short, soil erosion is the process of soil removal, while soil loss refers to the actual decrease in soil volume caused by erosion or other factors.
Soil erosion can result in soil loss, as well as reduced soil fertility, increased risk of flooding and landslides, and a decrease in biodiversity.
Round 2
Reviewer 2 Report
Comments and Suggestions for Authors
Thanks to the authors for implementing all of my suggestions.
Comments on the Quality of English LanguageMinor editing of English language required